# SatDiffMoE: A Mixture of Estimation Method for Satellite Image Super-resolution with Latent Diffusion Models

## Abstract

During the acquisition of satellite images, there is generally a trade-off between spatial resolution and temporal resolution (acquisition frequency) due to the on-board sensors of satellite imaging systems. High-resolution satellite images are very important for land crop monitoring, urban planning, wildfire management and a variety of applications. It is a significant yet challenging task to achieve high spatial-temporal resolution in satellite imaging. With the advent of diffusion models, we can now learn strong generative priors to generate realistic satellite images with high resolution, which can be utilized to promote the super-resolution task as well. In this work, we propose a novel diffusion-based fusion algorithm called **SatDiffMoE** that can take an arbitrary number of sequential low-resolution satellite images at the same location as inputs, and fuse them into one high-resolution reconstructed image with more fine details, by leveraging and fusing the complementary information from different time points. Our algorithm is highly flexible and allows training and inference on arbitrary number of low-resolution images. Experimental results show that our proposed SatDiffMoE method not only achieves superior performance for the satellite image super-resolution tasks on a variety of datasets, but also gets an improved computational efficiency with reduced model parameters, compared with previous methods.

## 1 Introduction

Satellite imaging is a very useful technique for monitoring the natural phenomena and human activities on the surfaces of the Earth. Lots of applications rely on satellite images such as crop monitoring, weather forecasting, urban planning, wildfire management and so on Khanna et al. (2024); Burke et al. (2021); Ayush et al. (2020; 2021); Beck et al. (2007); Wang et al. (2018a); M Rustowicz et al. (2019); Li et al. (2019). However, the acquisition of satellite images can be very expensive and the spatial and temporal resolution (the frequency that a satellite image is captured) may be limited due to the physical constraints of sensors Khanna et al. (2024). In addition, the high temporal resolution may come with trade-off in spatial resolution. Recent advance in satellite imaging technology enables us to capture the same area with a high-revisit frequency, but the spatial resolution is often limited. For instance, two Sentinel-2 satellites with resolutions from 10m to 60m can capture all land surfaces every five days Cornebise et al. (2022); Cong et al. (2022); Tarasiewicz et al. (2023); Van Etten et al. (2018), but to perform land crop monitoring or wildfire management, this resolution is not sufficient. On the other hand, some very high-resolution satellite images such as SPOT6 or WorldView can have resolution better than 1.5m Christie et al. (2018), but it is extremely difficult to collect those images for a large area, and these images cannot be captured as frequently as the Sentinel images. The limited temporal resolution severely limits downstream applications in urban planning, object detection, or continuous monitoring of crop or vegetation covers.

To solve the aforementioned challenges, super-resolution algorithms have been introduced to predict the high-resolution (HR) satellite images from a bunch of corresponding low-resolution (LR) satellite images Luo et al. (2018; 2023), so that to obtain more fine details. For example, given the low-resolution satellite images (10m) from Sentinel-2 of a specific location, the super-resolution algorithms are developed to predict the high-resolution (1.5m) SPOT6 satellite image of the same location at a specific time. Nevertheless, solving remote sensing super-resolution problems is still

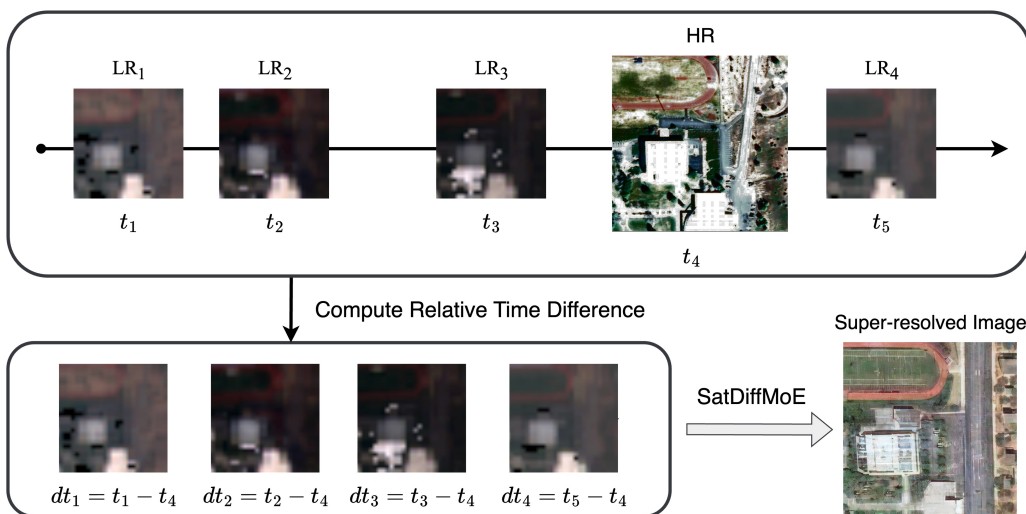

Figure 1: An overview of our proposed method SatDiffMoE.

an open problem. One challenge in remote sensing is that low-resolution and high-resolution images often come from different sensors, and may maintain very different image features due to the large imaging modality gap resulted from different sensors Luo et al. (2018). Moreover, since the low-resolution and high-resolution images are usually acquired at different time points as demonstrated in Fig. 1, a lot of atmospheric disturbance may pose additional challenges for modeling the sensor imaging process. Therefore, unlike the natural images, in the satellite images, the down-sampling process can be extremely difficult to model.

In order to tackle this challenging task of remote sensing super-resolution, considering that the acquisition of satellite images often comes with multiple revisits (a collection of satellite images at the same location but at different time stamps) or with multiple spectral bands, we hypothesize that fusing multiple low-resolution images with different spectrums or at different time stamps may provide complementary information to the model so as to benefit the super-resolution task. With such motivation, existing works Cornebise et al. (2022) have introduced a recursive fusion module that takes the concatenated LR images as inputs and applies a residual attention model for outputting the HR image. DiffusionSat Khanna et al. (2024) introduces a 3D ControlNet architecture that fuses LR images of different spectral bands to reconstruct the corresponding HR image. However, these works require a fixed number of LR images or require an absolute timestamp for each LR and HR, which is often not feasible and flexible at inference time in real practice because it is challenging to find a fixed amount of paired LR images for each HR image.

In this paper, we propose a novel diffusion-based method for solving the satellite image super-resolution problem as demonstrated in Fig. 1. Our contribution can be summarized as below:

- We propose a novel diffusion-based fusion algorithm for satellite image super-resolution that can take an arbitrary number of time series low-resolution (LR) satellite images as input, and fuse their complementary information to reconstruct high-resolution (HR) satellite images with more fine details.

- Specifically, we introduce a new mechanism to train a latent diffusion model using the paired LR and HR images (from the same location but with different time stamps), particularly being aware of the relative time difference between corresponding LR and HR images, to capture the time-aware mapping distribution of LR to HR images.

- At inference time, by leveraging the trained time-aware diffusion model, we propose a novel approach to fuse the information from time series LR images, by estimating the center of reverse sampling trajectories of different LR images using a perceptual distance metric, so as to align the semantics from various LR images for super-resolution task.

- We achieve the state-of-the-art performance on a variety of datasets for satellite image super-resolution. Moreover, our method demonstrates an improved computational efficiency with reduced model parameters compared with previous methods

## 2 BACKGROUND

**Latent diffusion models** Diffusion models consists of a forward process that gradually add noise to a clean image, and a reverse process that denoises the noisy images. The forward model is given by $x_t = x_{t-1} - \frac{\beta_t \Delta t}{2} x_{t-1} + \sqrt{\beta_t} \Delta t \omega$ where $\omega \in N(0,1)$. When we set $\Delta t \to 0$, the forward model becomes $dx_t = -\frac{1}{2}\beta_t x_t dt + \sqrt{\beta_t} d\omega_t$, which is a stochastic differential equation. The solution of this SDE is given by

$$dx_t = (-\frac{\beta(t)}{2} - \beta(t)\nabla_{x_t} \log p_t(x_t))dt + \sqrt{\beta(t)} d\overline{w}. \tag{1}$$

Thus, by training a neural network to learn the score function $\nabla_{x_t} \log p_t(x_t)$, one can start with noise and run the reverse SDE to obtain samples from the data distribution.

Latent diffusion models (LDM) Rombach et al. (2022) have been proposed for faster inference and training with a reduced computational burden. By applying an autoencoder to reduce data dimension, LDMs train the diffusion model in a compressed latent space, and then decode the latent code into signals. This method enables high-quality high-resolution image synthesis benefited from its compressed latent space, which is an ideal fit for satellite images due to the large image size. Nevertheless, it is still challenging to perform image restoration accurately with LDMs Song et al. (2024); Rombach et al. (2022). Various works have tried to extend LDMs for high-dimensional or high-resolution signal synthesis, such as video generation Blattmann et al. (2023); Yu et al. (2023); Ni et al. (2023); Ho et al. (2022); Ceylan et al. (2023); Martinez et al. (2021); Ruiz et al. (2023); Saharia et al. (2022); Voleti et al. (2022). However, few works apply LDMs for image fusion yet. It is an open problem to sample high-resolution images conditioning on multiple similar low-resolution images with LDMs.

**Single-image super-resolution** Single-image super-resolution (SISR) focuses on reconstructing the high-resolution image from one corresponding low-resolution image. In the era of deep learning, super-resolution problem has become a popular research question and many data-driven methods have been proposed. In past few years, state-of-the-art methods apply techniques such as Convolutional neural network (CNN), Generative Adversarial Network (GAN), and Transformers for restoring the high-resolution images Kingma & Welling (2013); Creswell et al. (2018); Haris et al. (2018); Wang et al. (2018b); Isola et al. (2017); Vaswani et al. (2017). These methods usually learn the mapping from low-resolution image to high-resolution image through a data-driven way using the pair data to train the neural network.

However, many of these methods output blurry or inaccurate reconstructions since they are learning a direct mapping between LR images and HR images without considering the distribution of possible HR images Song et al. (2024; 2023). Due to the ill-posedness of the super-resolution problem, there may exist multiple HR images corresponding to one single LR image. A direct regression-based approach may let the network learn an average of all possible HR images, which leads to blurry output. Diffusion models address this issue by learning a strong generative prior that can perform posterior sampling Chung et al. (2023) instead of direct regression. This sampling method outputs realistic images, and leads to better image perceptual quality. One line of work assumes the degradation operator is known, and focuses on inference-time posterior sampling with the diffusion prior without retraining Chung et al. (2023); Kawar et al. (2022); Wang et al. (2022); Song et al. (2024; 2021). The other line of work assume the degradation operator is unknown. They concat the LR image into the noise vector or conditional networks and then retrain or fine-tune the diffusion model Zhang et al. (2023); Khanna et al. (2024). Both lines of work show good ability to model complex high-resolution image distributions and outperform CNN approaches in image perceptual quality. However, all these works focus on obtaining HR images from a single LR image.

**Multi-image super-resolution** The goal of multi-image super-resolution is to combine the information from multiple LR images to reconstruct one HR image. In satellite imagery, different sensors

---

**Algorithm 1** SatDiffMoE: Satellite Image Fusion with Latent Diffusion Models

---

**Require:** $i$-$th$ low-resolution images $\text{LR}_i$, relative time difference $dt_i$, $i = 1, \ldots,$ N, Encoder $\mathcal{E}(\cdot)$,
Decoder $\mathcal{D}(\cdot)$, Score function $\boldsymbol{s}_\theta(\cdot, t)$, Pretrained LDM parameters $\beta_t, \bar{\alpha}_t, \eta, \delta$, Hyperparameter
$\lambda$ to control the fusion strength, $d(\cdot)$ the distance function.

$\boldsymbol{z}_T \sim \mathcal{N}(\boldsymbol{0}, \boldsymbol{I})$      ▷ Initial noise vector

**for** $t = T - 1, \ldots, 0$ **do**

    $\boldsymbol{\epsilon}_1 \sim \mathcal{N}(\boldsymbol{0}, \boldsymbol{I})$

    **for** all LR images **do**

    $\hat{\boldsymbol{\epsilon}}_{t+1}^i = \boldsymbol{s}_\theta(\boldsymbol{z}_{t+1}^i, t+1, \mathcal{E}(\text{LR}_i), dt_i)$      ▷ Compute the noise prediction

    $\hat{\boldsymbol{z}}_0^i(\boldsymbol{z}_{t+1}^i) = \frac{1}{\sqrt{\bar{\alpha}_{t+1}^i}}(\boldsymbol{z}_{t+1}^i - \sqrt{1 - \bar{\alpha}_{t+1}^i}\hat{\boldsymbol{\epsilon}}_{t+1}^i)$      ▷ Predict $\hat{\boldsymbol{z}}_0$ using Tweedie's formula

$\bar{\boldsymbol{z}}_0 \in \arg\min_{\boldsymbol{z}} \sum_{i=0}^N d(z, \hat{\boldsymbol{z}}_0^i(\boldsymbol{z}_{t+1}^i))$      ▷ Find the center of multiple $\hat{\boldsymbol{z}}_0$ by optimization

$\hat{\boldsymbol{z}}_0^i(\text{LR}_i) = (1 - \lambda)\hat{\boldsymbol{z}}_0^i(\boldsymbol{z}_{t+1}^i)) + \lambda\bar{\boldsymbol{z}}_0$

$\boldsymbol{z}_t^i = \sqrt{\bar{\alpha}_{t-1}}\hat{\boldsymbol{z}}_0^i(\text{LR}_i) + \sqrt{1 - \bar{\alpha}_{t-1} - \eta\delta^2}\hat{\boldsymbol{\epsilon}}_{t+1}^i + \eta\delta\boldsymbol{\epsilon}_1$      ▷ Update intermediate noisy samples

$\boldsymbol{x}_0^i = \mathcal{D}(\boldsymbol{z}_0^i)$      ▷ Output reconstructed image

---

have different resolutions. For instance, Sentinel-2 SITS has a resolution of 10-60m, but SPOT-6 or fMoW can have a resolution of less than 1.5m Khanna et al. (2024). Given sequential low-resolution images collected at the same location but different times, the hypothesis is that performing multi-image super-resolution is able to combine the information of LR images at multiple times to obtain a more accurate and higher-quality HR image. There are a couple of recent works in this venue Luo et al. (2018); Cornebise et al. (2022); Khanna et al. (2024). For example, Cornebise et al. trained a network with a traditional autoencoder architecture, but modify the encoder to incorporate multiple images as input Cornebise et al. (2022). HighRes-net Cornebise et al. (2022) adopted this idea to solve satellite image fusion problem with a network composed of an encoder, a recursive fusion network, and a decoder. However this approach does not include the temporal information of LR images. TR-MISR An et al. (2022) is a transformer-based fusion method for fusing an arbitrary LR input for super-resolution. The network consists of an encoder, a fusion module with self-attention, and a decoder. We observe that TR-MISR achieves impressive performance, but the it is still challenging to reconstruct realistic images with a larger resolution such as $512 \times 512$. There are also some works such as DDFM that solve the fusion problem with unconditional diffusion models, but not extending it to satellite image restoration tasks Zhao et al. (2023).

For satellite image restoration, DiffusionSat Khanna et al. (2024) proposed to train a 3D ControlNet on top of a fine-tuned latent diffusion model that leverages multispectral bands for reconstruction. However, at training, the 3D ControlNet requires the same number of low-resolution images for each paired high-resolution images. In addition, at inference-time, the number of low-resolution images used for reconstruction must be the same as the training set. Motivated by this, essentially different from all these previous works, we aim to propose a more flexible and robust fusion algorithm that can take an arbitrary number of low-resolution inputs with corresponding time information to generate the high-resolution satellite image.

## 3 METHODS

Instead of conditioning on multiple low-resolution images as a concatenated input Khanna et al. (2024), we propose a novel fusion algorithm: condition on **each image** and then fuse each score (the output of the conditional diffusion model) into a high-resolution image reconstruction. First, in order to embed the temporal information, we introduce a new method that trains a conditional diffusion model using one LR image and the relative time difference (between the input LR image and the target HR image) as inputs to reconstruct the HR image. Then we propose a novel inference-time algorithm that modifies the intermediate outputs in the reverse sampling procedure based on the assumption that HR reconstruction of LR images at the same location should look similar. Our method is illustrated in Fig. 2.

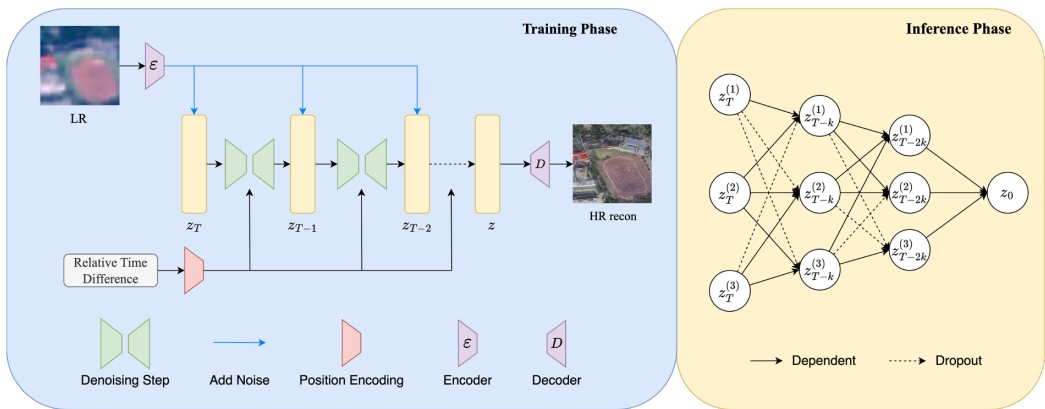

Figure 2: The framework of our proposed SatDiffMoE. In the training phase, we train a latent diffusion model for HR (high-resolution) image conditioning on a single LR (low-resolution) image and its relative time difference with the HR image. Then in the inference phase, we fuse the reverse sampling trajectories conditioning on each LR image of the same location. We can randomly select different trajectories for fusion, but output to a single image at the end.

**Training** We propose to reconstruct HR image from one LR image and its corresponding time difference with the target HR image. Firstly, we utilize Stable Diffusion Rombach et al. (2022), a pre-trained latent diffusion model for natural image generation, as the backbone model and fine tune this model on other image modalities. Let HR be the target high-resolution image we want to reconstruct, LR denote the low-resolution image collected at the same location as the HR image, and $dt$ denotes the relative time difference between LR and HR. Let $\mathcal{E}$ denotes the encoder, and $\mathcal{D}$ denotes the decoder in the LDM. Let $z_p = \mathcal{E}(\text{HR}_p)$, and $z_{LR} = \mathcal{E}(\text{LR})$, and $z_0\ \mathcal{E}(\text{HR})$ be the unknown ground truth image. The forward model of LDM is given by $\mathbf{z}_t = \sqrt{\alpha_t}z_0 + \sqrt{1 - \alpha_t}\epsilon$, where $\epsilon \in N(0, 1)$, and the goal of LDM is to predict $\epsilon$ when given $\mathbf{z}_t$ To train the conditional LDM (CLDM). We want our diffusion model to predict $\epsilon$ based on $\mathbf{z}_t$, and $z_{LR}$. However, the pretrained stable diffusion does not provide an additional layer to handle $z_{LR}$. Nevertheless, we can utilize the property of convolution to inject the LR image into training without introducing any additional parameters or changing the network architecture. Observe that the convolutional layer can take any resolution input as long it is divisible by k, where $k = 8$ in Stable Diffusion 1.5 Rombach et al. (2022). So we propose to enlarge each $z_t$ to be

$$\hat{\mathbf{z}}_t = \text{Concat}([\mathbf{z}_t, z_{lr}]) \tag{2}$$

Here the concatenation operation is happened on width dimension instead the channel dimension as a common practice in stable diffusion Rombach et al. (2022). The reason is convolutional layers need additional parameters to handle new channels, but not for the larger width. Then we want to fine-tune the network to denoise $\hat{\mathbf{z}}_t$. However, the network output $s_\theta(\hat{\mathbf{z}}_t)$ will also be the same size of $\hat{\mathbf{z}}_t$, which has a larger dimension that $z_0$. Recall that $\hat{\mathbf{z}}_t$ is an concatenation of the HR component: $\mathbf{z}_t$, and the LR component $z_{lr}$. So we propose to only optimize for the $HR$ component during fine tuning. One may wonder, if doing so, will the $LR$ component be useless in the fine tuning process? The answer is no, since the network takes both the HR and LR component for noise prediction. Even though we only take the HR component of the output, that part of output still has the information from LR component, so it will utilize that information for prediction.

Formally, the original training objective of LDM is given by

$$\text{argmin}_\theta \mathbb{E}[||\epsilon - s_\theta(\mathbf{z}_t)||] \tag{3}$$

now the new training objective becomes

$$\text{argmin}_\theta \mathbb{E}[||\epsilon - s_\theta(\hat{\mathbf{z}}_t)[:, :n//2, :]||] \tag{4}$$

where n is the second dimension of the latent code. In addition, since there is often a time mismatch between LR and HR. Then we can make an add-on to the training objective, by injecting the relative

time difference between LR and HR into the LDM training objective as additional conditional input. We make a clone of the time embedding network for the original Stable Diffusion model, feed $dt$ into the cloned embedding network, and then we add the output from the time embedding and the output from the $dt$ embedding together as an overall input to the diffusion model. We aim to reconstruct the same HR image regardless of what time LR is taken at, so we add this **relative time difference** embedding $dt$ to offset the time difference of LR and HR. Then the final training objective is given by:

$$\text{argmin}_\theta \mathbb{E}[||\epsilon - s_\theta(\hat{\boldsymbol{z}}_t, dt)[:, : n//2, :]||] \tag{5}$$

We call this method **DiffusionDrop**.

**Inference**  As discussed in the previous subsection, we want to estimate a single HR reconstruction based on $\text{LR}_i$ taken at different times. Hence we can assume that the outputs conditioning on different $\text{LR}_i$ should be aligned semantically since they are reconstructing the same HR image. Let $z_t(\text{LR}_i, dt_i)$ be a noisy sample at time $t$ during diffusion reverse sampling conditioning on $\text{LR}_i$ and $dt_i$. Based on the assumption above, we expect $\mathbb{E}[z_0|z_t(\text{LR}_i, dt_i)]$ to be similar for each $i_{th}$ LR. To achieve this, we propose a novel method that firstly finds the center $\bar{z}_0$ of the vectors of $\mathbb{E}[z_0|z_t(\text{LR}_i, dt_i)]$ for all i, and then updates each $z_t(\text{LR}_i, dt_i)$, so that $\mathbb{E}[z_0|z_t(\text{LR}_i, dt_i)]$ to be closer to the center than without updating.

We can find the desired center via optimization, where $d$ can be a specified distance function, and $N$ is the total number of low-resolution images. Note that $N$ can be different for different HR, which further demonstrates the flexibility of our method. For the distance function, we propose a novel approach that uses a convex combination of $l_2$ loss and LPIPS loss in order to prevent blurry outputs but still keep images close to each other.

$$\bar{z}_0 = \arg\min_x \sum_{i=1}^N d(x, \mathbb{E}[z_0|z_t(\text{LR}_i, dt_i)]) \tag{6}$$

$$d(x, x_i) = (1 - \alpha)\ell_2(x, x_i) + \alpha\text{LPIPS}(x, x_i) \tag{7}$$

where $\alpha$ is the weight for LPIPS loss. Note that for computational efficiency, when computing $\bar{z}_0$, we by design choose not to sum up every $\text{LR}_i$, but randomly sample a batch from the set of $\text{LR}_i$ to compute the summation. We observe in experiments this random batch selection strategy improves computational efficiency while not sacrificing performance,which may because of the redundancy information among LR images. Similarly for the computational efficiency, such optimization update is not performed on every time step, but on every k steps. In all of our experiments, we set $k = 5$, which we find from empirical study would suffice the fusion strength while reducing the computational cost.

After obtaining $\bar{z}_0$, we propose to update all intermediate samples from the $\text{LR}_i$ to be closer to $\bar{z}_0$. Recall that in DDIM reverse sampling Song et al. (2022), the reverse sampling can be decomposed by a clean image component and a noise component. We follow Chung et al. (2024)'s approach that only updates the clean image component, and leaves the noise component intact. Specifically, let $\lambda$ be a hyperparameter balancing the original clean image component and $\bar{z}_0$, we can update the new clean component as:

$$\hat{z}_0(\text{LR}_i, dt_i) = (1 - \lambda)\mathbb{E}[z_0|z_t(\text{LR}_i, dt_i)] + \lambda\bar{z}_0 \tag{8}$$

Therefore, the overall reverse sampling step can be written as

$$\boldsymbol{z}_{t-1} = \sqrt{\bar{\alpha}_{t-1}}\hat{z}_0(\text{LR}_i, dt_i) + \sqrt{1 - \bar{\alpha}_{t-1} - \eta\delta_t^2}\boldsymbol{s}_\theta(\boldsymbol{z}_t, \text{LR}_i, dt_i, t) + \eta\delta_t\boldsymbol{\epsilon}, \quad t = T, \dots, 0, \tag{9}$$

The pseudo-code of our proposed algorithm is demonstrated in Alg. 1.

## 4 Experiments

We try to answer the following questions in this section: (1) Can our proposed method achieve high-quality satellite image super-resolution results by fusing multiple time series low-resolution images? (2) Is our proposed fusion module effective? (3) Can our proposed method be more computationally efficient compared with previous methods? To study these questions, we benchmark the super-resolution performance on two widely used satellite image datasets: the fMoW dataset and the WorldStrat dataset.

| | Airport | Amusement Park | Car Dealership | Crop Field | Educational Institution | Electric Substation |
|---|---|---|---|---|---|---|
| WorldStrat | 0.723 | 0.732 | 0.747 | 0.738 | 0.733 | 0.736 |
| MSRResNet | 0.743 | 0.739 | 0.733 | 0.794 | 0.725 | 0.783 |
| DBPN | 0.763 | 0.740 | 0.728 | 0.783 | 0.726 | 0.750 |
| Pix2Pix | 0.621 | 0.652 | 0.652 | 0.645 | 0.647 | 0.643 |
| ControlNet | 0.625 | 0.653 | 0.648 | 0.650 | 0.658 | 0.644 |
| DiffusionSat | 0.623 | 0.647 | 0.637 | 0.649 | 0.652 | 0.639 |
| SatDiffMoE (Ours) | **0.579** | **0.626** | **0.600** | **0.608** | **0.612** | **0.606** |

Table 1: Comparison of the LPIPS metrics for super-resolution on the fMoW dataset for different categories. Best results are in bold.

## 4.1 DATASETS

**WorldStrat**    We take the paired LR-HR satellite image dataset from Cornebise et al. (2022). Each area of interest contains a single SPOT 6/7 high-resolution image with five bands. We take the RGB band of the SPOT6/7 satellite images, which has a GSD of 1.5 m/pixel. The low-resolution images are taken from the Sentinel-2 satellites consisting of 13 bands. We only pick the RGB band from them. For each area of interest, we have 16 paired low-resolution images taken at different time. The resolution ranges from 10 m/pixel to 60m per pixel. We crop the high-resolution image into 192x192 patches, and the low-resolution image into 63x63 patches.

**fMoW**    Function Map of the World (fMoW) Christie et al. (2018) consists of high-resolution (GSD 0.3m-1.5m) satellite images of a variety of categories such as airports, amusement parks, crop fields and so on. However, its temporal resolution is limited due to its high resolution. We use the metadata of timestamp for pairing low-resolution Sentinel-2 images. Using the dataset provided in Cong et al. (2022), we create a fMoWSentinel-fMoW-RGB dataset with paired Sentinel-2 (10m-60m GSD) and fMoW (0.3-1.5m GSD) images at each of the original fMoW-RGB locations. Then, we use the bounding box provided by the metadata to extract relevant areas, and then crop the high-resolution images to patches of 512x512. For each high-resolution fMoW image, we find the corresponding Sentinel-2 images of the same location. We only take the RGB band of Sentinel-2 images and then apply the same cropping method as that for fMoW-RGB images.

## 4.2 PERFORMANCE BENCHMARK

For super-resolution tasks on fMoW and WorldStrat datasets, we report perceptual quality metrics LPIPS to measure the perceptual similarity of the reconstructed image and the ground truth image, and FID to measure how realistic the reconstruction looks. We are also interested in the performance of downstream application of our reconstructed images.

**Zero-shot Classification with CLIP** In the fMoW dataset, we select the first 6 classes (airport, amusement parks, car dealership, crop field, educational institution, electric substation) for training and validation. To investigate how the reconstructed images perform for downstream applications, we compute the image embedding with a pretrained CLIP Radford et al. (2021) with ViT backbone, and a text embedding with the prompt "a satellite image of {class}". Then we compute the cosine similarity of each image with each class and select the class with the highest similarity. We denote this classification accuracy as "CA" score. More details can be found in the Appendix. We also report distortion metrics such as PSNR and SSIM for pixel-level similarity.

Note that in satellite images, LPIPS is a more relevant metrics here as it measures the perceptual similarity, as mentioned in Khanna et al. (2024).

| Method | WorldStrat | | fMoW | | |
|---|---|---|---|---|---|
| | LPIPS↓ | FID↓ | LPIPS↓ | FID↓ | CA↑ |
| WorldStrat | 0.481 | 139.3 | 0.736 | 426.7 | 15.83 |
| MSRResNet | 0.472 | 159.7 | 0.783 | 286.5 | 19.17 |
| DBPN | 0.475 | 122.6 | 0.750 | 278.2 | 15.17 |
| Pix2Pix | 0.427 | 93.90 | 0.643 | 196.3 | 22.67 |
| ControlNet | 0.580 | 108.0 | 0.644 | **102.3** | 39.00 |
| DiffusionSat | 0.561 | 92.97 | 0.638 | 102.9 | 34.50 |
| TR-MISR | **0.415** | 103.5 | 0.690 | 204.6 | 16.67 |
| SatDiffMoE (Ours) | 0.418 | **88.12** | **0.606** | 115.6 | **64.67** |

**Implementation Details** We evaluate our algorithms on WorldStrat and fMoW LR-HR datasets. For each AOI (Area of In-

Table 2: Comparison of LPIPS and FID metrics for super-resolution on WorldStrat dataset and fMoW. Best results are in bold. Second best results are underlined.

Figure 3: (Left) Super-resolution on fMoW dataset. (Right) Super-resolution on WorldStrat dataset.

terest) of the WorldStrat dataset, we resize each LR image to $192 \times 192$. We fix the prompt to be "Satellite Images" and compute the time difference between the low-resolution image and high-resolution image. Then, we add the time difference embedding network to the stable diffusion 1.2 and then fine-tune the stable diffusion 1.2 on the low-resolution and high-resolution pair of the WorldStrat dataset conditioning on the time difference. For each high-resolution image, we randomly select a low-resolution image from its 16 corresponding low-resolution images for training. After that, we get a latent diffusion model that takes a low-resolution image and the relative time difference as input and output the predicted high-resolution image.

For each AOI of the fMoW dataset, we resize each LR image to $512 \times 512$ to align with the size of high-resolution image.

During inference, for both datsets we use 50 NFEs, and then perform optimization every 5 steps. More implementation details can be found in the supplementary materials.

For WorldStrat, we evaluate our algorithm on first 1000 images in the validation set. For fMoW, we select the first 100 images in the validation set from 6 classes: Airport, Amusement Pak, Car Dealership, Crop Field, Educational Institution, Electric Substation for evaluation.

| Method | WorldStrat | | fMoW | |
|---|---|---|---|---|
| | Parameters (M) | Training Iterations | Parameters (M) | Training Iterations |
| ControlNet | 1427 | 12500 | 1427 | 21400 |
| DiffusionSat | 1428 | 12500 | 1428 | 20000 |
| Ours | **1068** | **800** | **1068** | **4400** |

Table 3: Number of parameters and memory required.

| RelativeTimeDiff | Fusion | PSNR ↑ | SSIM ↑ | LPIPS ↓ | FID ↓ |
|---|---|---|---|---|---|
| | | 12.52 | 0.122 | 0.565 | 102.3 |
| ✓ | | 15.28 | 0.272 | 0.496 | 105.1 |
| ✓ | ✓ | 17.40 | 0.396 | 0.418 | 88.12 |

Table 4: Evaluating the Effectiveness of Relative Rime Difference and Fusion.

**MultiSpectral Training**  In addition to the RGB bands in the Sentinel-2 LR images, we can access additional bands for conditioning. Following the work Khanna et al. (2024), we take the SWIR and NIR bands of the Sentinel-2 images for conditioning. We apply our a modified "Diffusiondrop" training mechanism with the modified multi-spectral inputs. We apply an additional convolutional layer to the additional bands and then add the output to the concatenation of the code of LR and HR images. Since multispectral results are not always available, so we do not report quantitative results. However, when the RGB LR bands are corrupted, we can leverage the multispectral inputs for conditioning. Figure. 4 demonstrates we can reconstruct very realistic images on the fMoW validation dataset with our SatDiffMoE method with multispectral inputs.

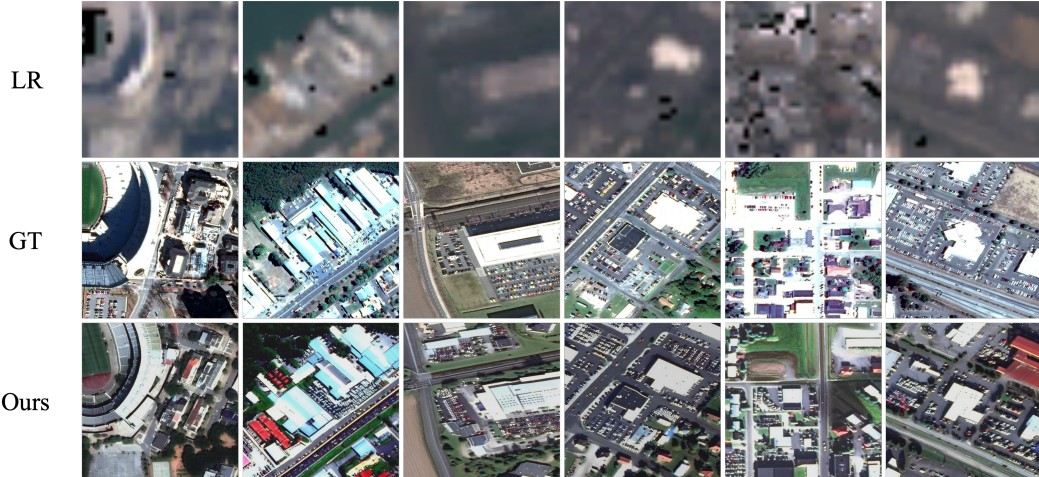

LR

GT

Ours

Figure 4: Super-resolution results of SatDiffMoE with multispectral low-resolution image as the input. We only display the first three spectrums of the LR image that is closest to the HR image in time

**Baselines** We consider six state-of-the-art baselines, WorldStrat Cornebise et al. (2022), MSR-ResNet Wang et al. (2018b), DBPN Haris et al. (2018), Pix2Pix Isola et al. (2017), ControlNet Zhang et al. (2023), and DiffusionSat Khanna et al. (2024). Both ControlNet and DiffusionSat are diffusion-based methods that takes LR as an additional condition for fine-tuned stable diffusion to predict the HR image. Both MSRResNet and DBPN are CNN-based methods that directly map the low-resolution image to the high-resolution image, and Pix2Pix is a GAN-based method. Both WorldStrat and DiffusionSat are fusion-based methods that fuse multiple low-resolution input to predict the HR image, while others take single low-resolution input and predict the HR image.

**Results and Discussions** The LPIPS scores on six selected class in fMoW dataset are reported in Table. 1. We also report LPIPS and FID scores in Table 2 on both fMoW and WorldStrat datasets compared to the six baselines mentioned before. We observe that our algorithm largely achieves better or comparable performance in perceptual quality. Our method achieves state-of-the-art LPIPs and CA score compared to all baselines and comparable FID scores. More importantly, our method also demonstrates **significant** improvement over other methods on downstream zero-shot classification accuracy of satellite images, in which we beat the second-best by 25.67. CNN-based baselines tend to perform poorly in perceptual quality, resulting in sub-par FID scores compared to diffusion-based methods. While ControlNet may show slightly better FID scores, but the LPIPs score is significantly worse than ours. We also show reasonable PSNR and SSIM scores, as demonstrated in the Appendix. Qualitatively, as demonstrated in Fig. 3, we also demonstrate that our method is able to capture fine-grained details. Compared to baselines, we are able to reconstruct both realistic images and accurate details.

**Computational Efficiency** We observe that the training phase of our proposed method requires much fewer parameters and iterations to converge than ControlNet and DiffusionSat, while achieving comparable or better reconstruction performance. We report the number of parameters and number of iterations of training in Table. 3. We train each model until the FID stops improving. Our method converges significantly (5-15 times) faster than ControlNet and DiffusionSat on both datasets. We also do not observe the "sudden convergence phenomena" in our training which is reported in ControlNet, which implies that our training may be more stable. Our method also exhibits better computational efficiency compared to ControlNet. As demonstrated in Table. 3, our algorithm requires significantly less training time than ControlNet.

### 4.3 ABLATION STUDIES

We want to study the impact of the fusion module and the relative time difference embedding on the reconstruction quality. We are also interested in the number of images for fusion. Intuitively, with

| Number of images for fusion | 2 | 4 | 6 | 8 | 10 | 12 | 14 | 16 |
|---|---|---|---|---|---|---|---|---|
| LPIPS | 0.41124 | 0.3967 | 0.3920 | 0.3891 | 0.3877 | 0.3874 | 0.3867 | 0.3866 |
| Inference Time (s) | 16.00 | 17.11 | 18.82 | 20.64 | 22.45 | 24.25 | 25.99 | 27.81 |

Table 5: Ablation study: number of images for fusion

| Method | DiffusionDrop (Ours) | ControlNet | Additional Layers |
|---|---|---|---|
| LPIPS | **0.610** | 0.624 | 0.634 |
| Iterations of Convergence | **4400** | 21400 | 20600 |

Table 6: Effect of different conditioning mechanisms on the fMoW airport validation set

more LR images provided for fusion, we have more complementary details and can achieve better perceptual quality.

**Effectiveness of relative time embedding and fusion**   We demonstrate the performance of our method without RelativeTimeDiff and Fusion, our method with RelativeTimeDiff, and our full method on the WorldStrat validation dataset in Table. 4. We found that both modules have significant positive impact to the perceptual quality (LPIPS score), while the fusion module improves both the LPIPS score and the FID score. On the other hand, we also find improvement in the distortion metrics when adding both modules.

**Impact of number of images on fusion**   We present the effect of increasing number of image for fusion on the reconstruction quality on the first 100 images in the validation set of WorldStrat dataset in Table. 5. We observe that the reconstruction quality improves significantly when adding more images for fusion when there are few images for fusion (i.e. fewer than 4), but the performance almost saturates when the number of images exceeds 10, and will trade-off between image quality and inference time. This observation validates our hypothesis that fusing information from multiple LR images improves the reconstruction quality and there exists a diminishing return on the number of LRs.

**Effectiveness of DiffusionDrop**   We fine tune the stable diffusion model with DiffusionDrop (Ours), a ControlNet, and additional convolutional channels for handling input condition (with the same fMoW training data), and then tested the reconstruction on the validation set of fMoW airport class. We observed that in addition to a fast convergence as demonstrated in Table. 3, Our DiffusionDrop method also follows the LR image better than both the ControlNet and Additional layers method. We observe a better LPIPS score with the proposed method as demonstrated in Table. 6

## 5 CONCLUSION

In this work, we present "SatDiffMoE", a novel framework for satellite image super-resolution with latent diffusion models. We first present a novel training mechanism that conditions on relative time difference of LR images and HR images. Then, we propose a novel inference-time algorithm that fuses the reverse sampling trajectory from inputs of different LRs at the same location but different times. Our method is highly flexible that can adapt to an arbitrary number of low-resolution inputs at test-time and requires fewer parameters than diffusion-based counterparts. One of our limitation is we do not impose physical measurement constraint in our reconstruction process, which we will leave as future work.

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

# A APPENDIX

## A.1 IMPLEMENTATION DETAILS

**Data Preprocessing    WorldStrat** We obtain the RGB satellite images provided by Cornebise et al. (2022). We follow the preprocessing steps given by Cornebise et al. (2022), that crop the HR images into patches of $192 \times 192$ and LR images into corresponding pathes of the HR images of size $63 \times 63$. For each HR image, there are 16 corresponding LR images of different time. We extract the RGB band of the HR images and the RGB band of every LR images. We then resize LR image into the size of $192 \times 192$. We use the same training and validation split provided by Cornebise et al. (2022) and then form a training and validation set. We then extract the timestamp from the metadata and compute the $dt_i$ for each $LR_i$.

**fMoW** We obtain the high-resolution images from Christie et al. (2018), and the paired Sentinel-2 images from Cong et al. (2022). We first identify the area of interest on the HR images as given by Christie et al. (2018), and then crop out other areas. Then we crop the corresponding LR images following the pre-processing steps given by Cong et al. (2022). We crop the HR images into patches of $512 \times 512$, and align LR images into patches based on each HR image patch. Then we resize each LR image into the size of $512 \times 512$ in accordance with the HR image. We consider 6 categories: airport, amusement parks, car dealership, crop field, educational institution, electric substation for training and testing. For training, we filter out HR images that do not have a corresponding LR, and those do not have three channels. We take the same training and validation split from Christie et al. (2018). When training, we consider all images from Christie et al. (2018), and when testing, we pick the first 100 images from each selected category of the validation set of Christie et al. (2018). We also extract $dt_i$ from the metadata of fmow provided by Christie et al. (2018), and the metadata of Sentinel-2 data provided by Cong et al. (2022).

**Model Training** We take the pretrained checkpoint (SD1.2) provided by Rombach et al. (2022), and then fine tune on the processed Christie et al. (2018) and Cornebise et al. (2022) datasets. We rescale every LR image and HR image to the scale of [0,1]. Then, we use a learning rate $1e^{-5}$, and a batch size of 4 for both datasets. We stop training when the FID of sampled images stops improving. For Cornebise et al. (2022) dataset, we only train 800 iterations (partly due to the small dataset size). For Christie et al. (2018) dataset, we train 4400 iterations. We then take the model for downstream inference tasks. For WorldStrat dataset, we use the prompt "Satellite images" for training.

**Model Inferencing** We use 50 DDIM steps with $\eta = 0$ for inferencing. We perform optimization every 5 steps for computational efficiency, otherwise, we just perform conditional sampling. We set $\lambda = 0.1$ and $\alpha = 0.2$ for both datasets.

**CLIP zero-shot classification** We use the pretrained CLIP model to compute the image embedding and text embedding. For each class of fMoW, we take the reconstructed images from each methods and compute its CLIP embedding using the ViT-B/32 model, which gives an embedding of shape (1,512). Then we use the text encoder to compute the embedding, which gives a shape of (77, 512). The prompt for each class is given by "a satellite image of {class} from an overhead view". Then we commpute the cosine similarity of the image embedding and the text embeddings, which takes the mean of the text embedding over dimension 0. Then we pick the class with the highest cosine similarity.

## A.2 MORE ABLATION STUDIES

**Effect of LPIPS weight $\alpha$ and optimization weight $\lambda$** There are two hyper-parameters in our inference-time algorithm, and that is the weight for LPIPS distance $\alpha$ v.s. L2 distance, and the weight $\lambda$ for balancing the original predicted clean image component and the one after optimization. We expect the perceptual quality of reconstructed images to improve when we increase $\alpha$ from 0. We also expect the reconstruction quality to improve when $\lambda$ increases from 0 since the weight of fusion increases. In Fig.5, we present the LPIPS score on 100 samples on the WorldStrat dataset with varying $\alpha$ and $\lambda$. We find that LPIPS score improves when both $\alpha$ and $\lambda$ increase from 0. Then, performance converges as $\alpha$ keeps increasing, and marginally degrades as $\lambda$ continues to increase. We observe that generally, the performance of our algorithm is insensitive to hyperparameter change.

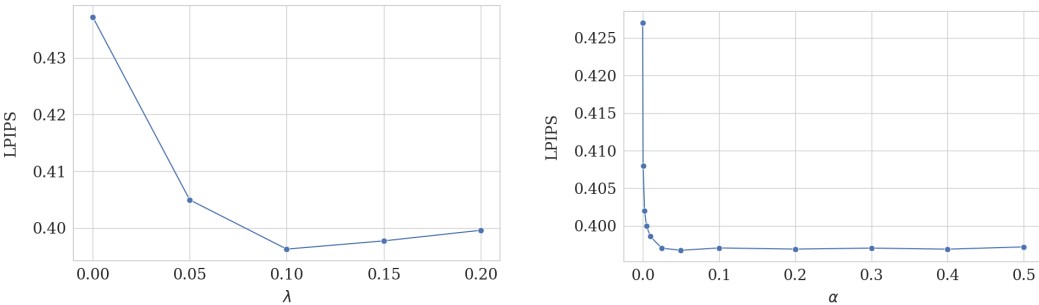

Figure 5: (Left) Ablation study on optimization weight $\lambda$. (Right) Ablation study on LPIPS weight $\alpha$.

### A.3 IMPLEMENTATION DETAILS OF BASELINES

**WorldStrat**   We follow the original codebase of Cornebise et al. (2022), where we train the High-ResNet model on both Christie et al. (2018) and Cornebise et al. (2022) datasets. We tune the hyperparameters of the loss function based on our validation set. We stop training when the validation performance converges. On Cornebise et al. (2022) we train 125000 iterations with a batch size of 32, and on Christie et al. (2018) we take 100000 iterations with a batch size of 32.

**MSRResNet**   We follow the original codebase of Wang et al. (2018b), where we train the MSR-ResNet model on both Christie et al. (2018) and Cornebise et al. (2022) datasets. During training, we randomly pick a LR image and its paired HR image. We tune the hyperparameters of the loss function based on validation set performance. We train for 160000 iterations for both Cornebise et al. (2022) and Christie et al. (2018) datasets with a batch size of 16.

**DBPN**   We follow the original codebase of Haris et al. (2018), where we train the MSRResNet model on both Christie et al. (2018) and Cornebise et al. (2022) datasets. During training, we randomly pick a LR image and its paired HR image. We tune the hyperparameters of the loss function based on validation set performance. We train for 100 epochs for both Cornebise et al. (2022) and Christie et al. (2018) datasets with a batch size of 16.

**Pix2Pix**   We follow the original codebase of Isola et al. (2017), where we train the MSRResNet model on both Christie et al. (2018) and Cornebise et al. (2022) datasets. During training, we randomly pick a LR image and its paired HR image. We tune the hyperparameters of the loss function based on validation set performance. We train for 30 epochs for Christie et al. (2018) and 100 epochs for Cornebise et al. (2022) with a batch size of 16.

**ControlNet**   We follow the original codebase of Zhang et al. (2023). During training, we randomly pick a LR image and its paired HR image. We tune the hyperparameters of the loss function based on validation set performance. We train for 12500 iterations with a batch size of 16 for Cornebise et al. (2022), and 21500 iterations with a batch size of 16 for Christie et al. (2018).

**DiffusionSat**   We implemented the 3D ControlNet architecture as mentioned in Khanna et al. (2024). Then, we take the RGB band and the SWIR, NIR band from LR image for training 3D ControlNet. We tune the hyperparameters of the loss function based on validation set performance. We train for 12500 iterations with a batch size of 16 for Cornebise et al. (2022), and 20000 iterations with a batch size of 16 for Christie et al. (2018). We observe that further training worsened FID scores on both datasets.

**TR-MISR**   We use the repo from https://github.com/Suanmd/TR-MISR/. We set the learning rate to be $1e^{-4}$ which we observe to give the best performance, and then we use the l-2 loss. We train for 100 epochs for both datasets with early stopping if the validation loss stops decreasing.

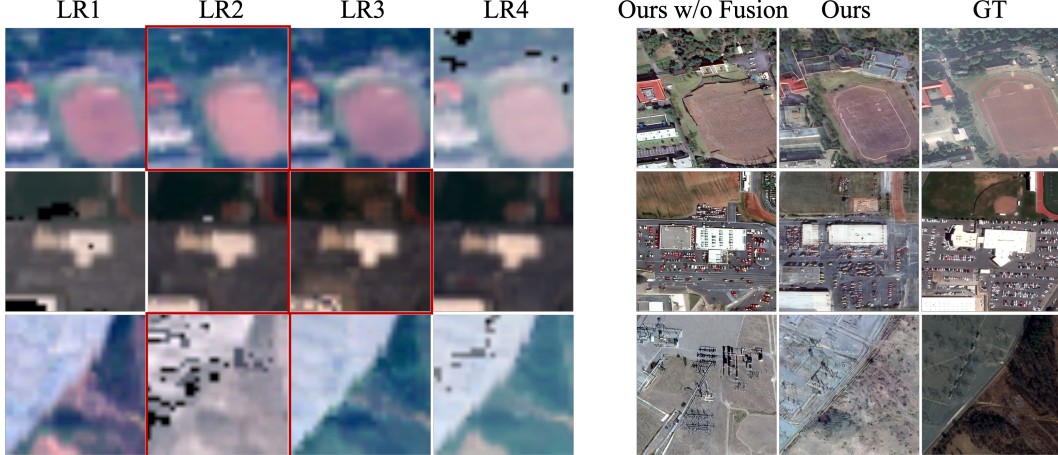

LR1  LR2  LR3  LR4  Ours w/o Fusion  Ours  GT

Figure 6: Super-resolution results of SatDiffMoE with paired low-resolution image as the input.

## A.4 MORE RESULTS

We present additional results on reconstruction for paired LR and HR (taken at the same time) in Fig. 6. Notice that we can reconstruct accurate details in this setting. We report additional results on unconditional generation and conditioning on $dt_i$ as demonstrated in Fig.7, and Fig.8. We observe that we can generate realistic satellite images. Conditioning on $dt_i$ makes semantic changes in the image and can be applied to tasks such as cloud removal. We also report the PSNR and SSIM metrics in Table 7. The error bars are presented in Table 8.

| Method | WorldStrat | | fMoW | |
|---|---|---|---|---|
| | PSNR↑ | SSIM↑ | PSNR↑ | SSIM↑ |
| WorldStrat Cornebise et al. (2022) | 17.98 | 0.396 | **13.42** | **0.443** |
| MSRResNet Wang et al. (2018b) | **19.81** | **0.512** | 13.01 | 0.290 |
| DBPN Haris et al. (2018) | 19.17 | 0.471 | 11.90 | 0.268 |
| Pix2Pix Isola et al. (2017) | 19.76 | 0.448 | 12.21 | 0.180 |
| ControlNet Zhang et al. (2023) | 11.89 | 0.113 | 10.82 | 0.117 |
| DiffusionSat Khanna et al. (2024) | 12.34 | 0.133 | 10.63 | 0.109 |
| SatDiffMoE (Ours) | 17.40 | 0.396 | 11.96 | 0.172 |

Table 7: Comparison of PSNR and SSIM metrics for super-resolution on WorldStrat dataset and fMoW. Best results are in bold. Second best results are underlined.

| Method | WorldStrat | MSRResNet | DBPN | Pix2Pix | ControlNet | DiffusionSat | SatDiffMoE(Ours) |
|---|---|---|---|---|---|---|---|
| WorldStrat | 0.081 | 0.077 | 0.079 | 0.069 | 0.079 | 0.091 | 0.076 |
| fMoW | 0.092 | 0.081 | 0.052 | 0.045 | 0.034 | 0.034 | 0.044 |

Table 8: Standard deviation of the quantitative metric LPIPS presented in Table 2 for super-resolution on fMoW and WorldStrat dataset.

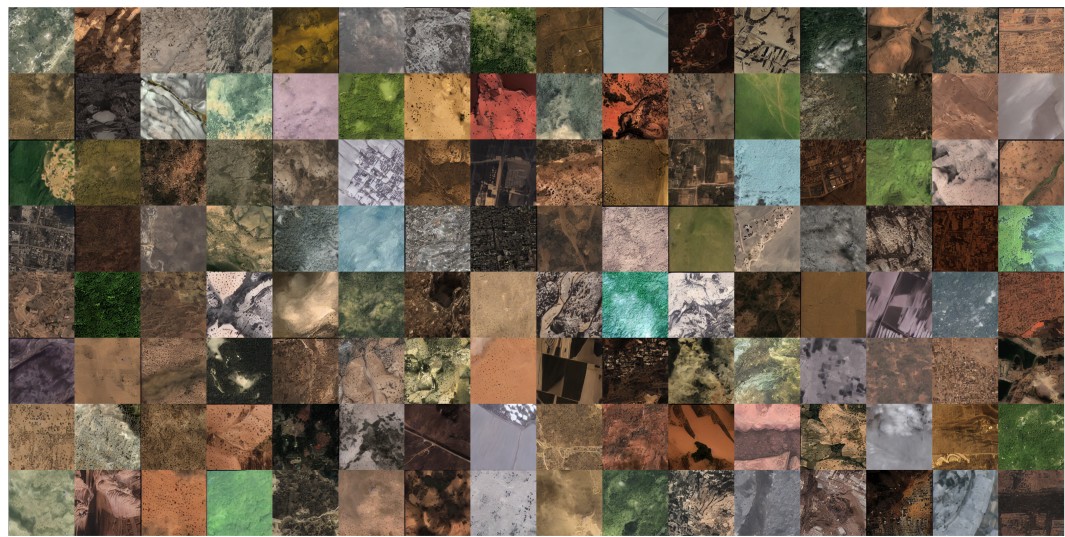

Figure 7: Generated high resolution samples from our unconditional model.

LR

GT

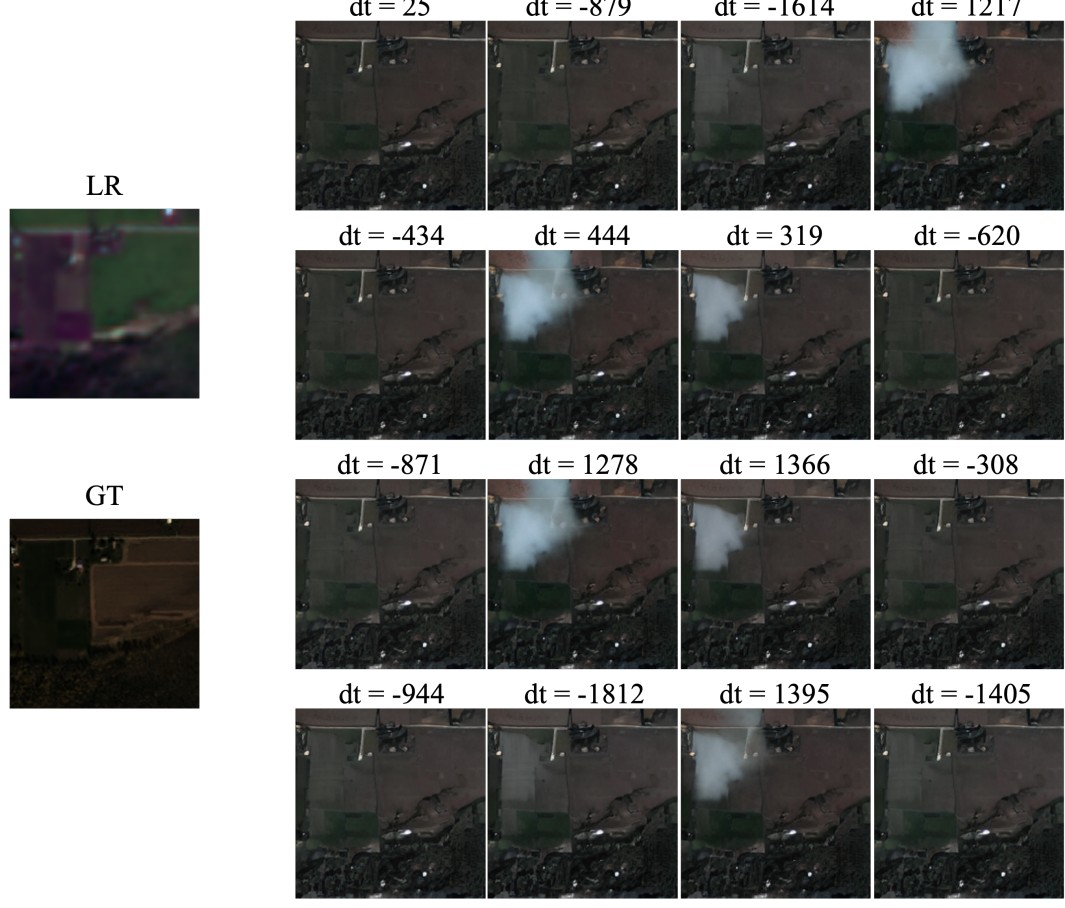

Figure 8: Here we consider the same LR image, and vary the relative time difference to generate different conditional samples.

