# OpenReview forum: "SatDiffMoE: A Mixture of Estimation Method for Satellite Image Super-resolution with Latent Diffusion Models"
_ICLR.cc/2025/Conference — ICLR 2025 Conference Withdrawn Submission_

### Official Review · Reviewer_Snpo · 2024-10-29

**Soundness:** 3
**Presentation:** 2
**Contribution:** 2
**Rating:** 5
**Confidence:** 4

**Summary:**

This paper proposes a new fusion super-resolution method based on the diffusion model. Its feature is that it allows any number of low-resolution images to be used as input to reconstruct a high-resolution image. To enhance the mapping difference between the low-resolution images at different timestamps and the corresponding high-resolution images, it constructs the relative time difference and uses it as an additional condition input. Another contribution is that during the inference phase, the trajectories corresponding to each low-resolution image can be randomly selected for fusion and finally output to a single high-resolution image. Experiments show the advantages of the proposed method.

**Strengths:**

+ This method is very flexible and can accept sequences with different numbers of low-resolution images for fusion. This advantage is attributed to its specific fusion strategy for sampling trajectories.
+ The definition and conditional input of the relative time difference is also interesting, which can help the network capture the time-aware mapping distribution of low-resolution to high-resolution images.
+ Experimental results show that the reconstruction results are relatively better than competitors in terms of rationality and naturalness.

**Weaknesses:**

- A key operation is F, which masks out the HR component from the score function. However, this paper does not explain how F was designed. This makes the description of the relevant methodology unclear.
- There is also some disagreement about the definition of the relative time difference. In Figure 1, time t4 corresponds to the high-resolution image, and dti represents the difference between the low-resolution image at time ti and the high-resolution image. However, the definition of dt4 is obviously different from the others, which is not rigorous in expression.
- I am concerned about the significance of this technology. On the one hand, images captured at different times may show changes in surface cover, so is it appropriate to use a high-resolution image at one of these moments as a reference? On the other hand, the experiments also confirmed my concern that the reconstructed results of the proposed method, although better than those of other methods, still show considerable differences with GT. This difference is not only reflected in the intensity distribution, but also includes changes in some ground objects. This makes me concerned about whether the reconstructed high-resolution results are credible and applicable.
- Sequence fusion with multiple low-resolution images seems to only appear in ablation studies. Is the comparative experiment just a single image super-resolution?
- Could this method be extended to consider the possibility of parallax between low-resolution images? In reality, images captured at different times theoretically have geometric differences
- There are some typos. For example, the last line of page 4 is missing a period.

**Questions:**

See weaknesses.

---

> ### Comment · Reviewer_ZHen · 2024-11-26
>
> No feedback was received from the authors. I keep the rating unchanged.

---

### Official Review · Reviewer_XAtp · 2024-11-01

**Soundness:** 2
**Presentation:** 1
**Contribution:** 2
**Rating:** 5
**Confidence:** 5

**Summary:**

This paper introduces a super-resolution latent diffusion model based on finetuning SD1.2 model, to fuse and SR and remote sensing images. To adapt fusing different LRs at different times, the authors proposed DiffusionDrop and optimized-based diffusion to fuse diffusion trajectories. Results on some remote sensing image SR datasets show its effectiveness.

**Strengths:**

1. Propose DiffusionDrop and use previous $z_0$ optimization to fuse LRs to HR.

2. Good results on different satellite image dataset.

**Weaknesses:**

Some typos:

1. At line 169 in Algo. 1, the comment ''Compute the score'' actually should be ''noise'', since you're using SD1.2 model, which is trained on noise-prediction objective.
2. At line 103, there should exist an semicolon.
3. At line 166, there should exist the encoder to encode the LR images.

Weakness:

1. Some key components like $F$ in Eq. (2), finding the center ($\bar z$) is too vague, hindering reader's understanding.

2. In inference, the authors claim that finding the center ($\bar z_0$) of $z_i$-s are based on Chung et al (2024), but actually, in this refered paper, when optimzing the $z_t$ (or says diffusion trajectory), CG steps should be taken to optimize with the help of the degraded operator ($\mathbf A$). It seems that you did not have the operator $\mathbf A$ and did not include the description of how to take the $\arg \min$.

3. According to your description in inference stage, finding the center of predicted $\hat z_0$ should include all LRs which is $LR_i, i\in \{0,\dots, N-1\}$ (if you have $N$ LRs when inference). This means in Algo. 1, there should be **two** for-loops, one for diffusion timesteps and one for $N$ LR samples. Did I misunderstand?

4. In the experiments, the authors compared with ControlNet, but only training it with only one LR and HR pair. This comparison is unfair, since the proposed method uses multiple LRs.

5. As shown in your Eq. (2), inputs should be encoded LRs and HR, but in Fig. 2, there is only LR and do not have $F$.

---

This paper propose some techniques to finetune the pretrained SD model to adapt on satellite SR task with various number of LR images, but the presentation is too poor to understand its contributions.

I believe the proposed method's SR performance can beat previous methods, such as previous regressive models and diffusion models, but the presentation and writing is still an important part. The writting makes this paper can not be accepted by ICLR.

Now, I rate this paper to 5 (marginally below the acceptance threshold).

**Questions:**

1. What's the $F$ in Eq. (2), I did not find its implementaion in the rest of paper. Only description like ''mask out the HR component'' is too vague.

2. Please provide 1) the details of used datasets, including train/validation/test ratios, total number of images etc; 2) the computation resources to finetuning the SD1.2 model.

---

> ### Comment · Reviewer_XAtp · 2024-11-26
>
> No feedback. I keep the rating unchanged.

---

### Official Review · Reviewer_DVW4 · 2024-11-02

**Soundness:** 2
**Presentation:** 2
**Contribution:** 2
**Rating:** 3
**Confidence:** 4

**Summary:**

The authors propose a novel diffusion-based fusion algorithm called SatDiffMoE, which accepts an arbitrary number of sequential low-resolution satellite images to generate a high-resolution reconstructed image. During sampling, SatDiffMoE first obtains the center of the intermediate sampling results from different low-resolution conditional images and then weights this center along with the intermediate results to produce the next sampling. Experimental results demonstrate that SatDiffMoE can synthesize high-quality images.

**Strengths:**

(1) It is novel that using the timestamp differences as embedding prompts.
(2) The sampling strategy effectively combines the intermediate sampling results from different low-resolution conditional images, enabling the fusion of an arbitrary number of low-resolution images.
(3) SatDiffMoE can synthesize high-quality images.

**Weaknesses:**

(1) The mechanism of DiffusionDrop is unclear and needs further clarification.
(2) The content restoration of high-resolution images generated by SatDiffMoE lacks competitiveness.
(3) The literature review is insufficient.
(4) The comparison methods are either single-image super-resolution approaches or designed for synthetic data, lacking comparisons with multi-image super-resolution methods, especially those tailored for remote sensing imagery.

**Questions:**

(1) How does the operator $F$ function in Eq. (2), and why is it necessary to use $F$ to mask out the HR component from the score? Under certain conditions, the score function $s_\theta(\cdot, t)$ can be interpreted as modeling the conditional probability $p(HR_t | HR_{t-1}, LR)$. Thus, once the available score function is obtained from the training dataset, it can gradually sample the HR from Gaussian noise at inference stage.
(2) Referring to the background on multi-image super-resolution, the rank 1 method [1] on the PROBA-V dataset (the mentioned HighRes-net ranked 6th) can take an arbitrary number of low-resolution inputs to generate a high-resolution satellite image. However, the authors did not address this point, which is closely related to the contributions of this paper, specifically regarding the concept of "arbitrary number."
(3) Referring to Line 133, "However, few works apply LDMs for image fusion yet", actually, there are lots of DM-based image fusion mothed have been developed last 3 years.
(4) As a method to obtain accurate HR images rather than synthetic HR images, I am more focused on image content restoration quality, measured by PSNR and SSIM, rather than the quality of image generation and perceptual quality. If the goal is to generate high-quality satellite data, why not directly train a SAT Stable Diffusion model instead of using a fusion model? Compared to the regression-based model MSRResNet (Table 7), the restoration quality of the proposed method is not particularly outstanding. Additionally, some of the comparative methods are not specifically designed for multi-image super-resolution; rather, they are either single-image super-resolution methods or intended for synthetic data.

[1]An, Tai, et al. "TR-MISR: Multiimage super-resolution based on feature fusion with transformers." _IEEE Journal of Selected Topics in Applied Earth Observations and Remote Sensing_ 15 (2022): 1373-1388.

---

> ### Comment · Reviewer_DVW4 · 2024-11-26
> **Reback**
>
> No feedback from the authors. I will maintain my score

---

### Official Review · Reviewer_ZHen · 2024-11-16

**Soundness:** 2
**Presentation:** 2
**Contribution:** 2
**Rating:** 5
**Confidence:** 3

**Summary:**

The paper llooks sound based on the idea "Super-resolution with Latent Diffusion Models for Satellite Image" This method introduces SatDiffMoE, a novel diffusion-based algorithm that fuses sequential low-resolution images into high-resolution outputs using complementary time-point data. The method demonstrates flexibility, superior performance, and computational efficiency on super-resolution tasks across various datasets.

**Strengths:**

1. This method is highly flexible, capable of handling sequences with varying numbers of low-resolution images for fusion
2. Its adaptability stems from a specific fusion strategy for sampling trajectories, enhanced by the novel use of timestamp differences as an embedding prompt to guide the fusion process effectively.
3. This strategy harmonizes intermediate outputs across varying low-resolution inputs, enabling a flexible and efficient fusion process independent of the image count.

**Weaknesses:**

1. Limited comparison with SOTA methods
2 Literature review is not strong, problem statement
3. The use of diffusion models introduces a dependency on the quality of the generative priors
4. In real-world conditions (e.g., different climates or satellite systems), the generalizability of the results might be a concern.

**Questions:**

1. Why the proposed model was designed for satellite image , why not for real image datasets like Urban100
2. Complexity of the season still huge, if the comparison will be made with weight methods

2. Address the weakness motioned in section.

---

### Note · Authors · 2025-01-04

**Comment:**

We are thankful to the constructive discussion. However, there are issues unsolved in the discussion. Therefore, we decide to withdraw this submission.

**Withdrawal Confirmation:**

I have read and agree with the venue's withdrawal policy on behalf of myself and my co-authors.